# An Ant-Mimicking Jumping Spider Achieves Higher Predation Probability with Lower Success Rate When Exposed to Ethanol

**DOI:** 10.3390/insects13111009

**Published:** 2022-11-01

**Authors:** Guocheng Yu, Zichang Li, Yao Zhao, Jie Liu, Yu Peng

**Affiliations:** 1Hubei Key Laboratory of Regional Development and Environmental Response, College of Resources and Environmental Science, Hubei University, Wuhan 430062, China; 2State Key Laboratory of Biocatalysis and Enzyme Engineering and Centre for Behavioural Ecology and Evolution, School of Life Sciences, Hubei University, Wuhan 430062, China

**Keywords:** ETOH, salticids, ETOH preference, prey capture efficiency, predation probability

## Abstract

**Simple Summary:**

Ethanol is a non-negligible factor which affects many animals’ behavior in nature. Jumping spiders can be found in almost every type of environment, and thus can easily make direct contact with ETOH. However, it remains unclear how ETOH affects jumping spiders. In this study, we tested the ETOH preference of jumping spiders. Jumping spiders show a significant preference at a medium concentration and avoidance at a high concentration of ETOH. We also tested the potential benefits jumping spiders may obtain from choosing ETOH in terms of predation probability and prey capture efficiency. We found that although ETOH would decrease the success rate of the first attack, the predation probability would be higher due to the fact that one type of their prey, the fruit fly, showed significant preference for ETOH. Our findings suggest that spiders may have evolved to use ethanol as a signal of the presence of food resources, and also provide a new direction for ETOH-invertebrates interactions.

**Abstract:**

Ethanol (ETOH) affects many animals’ behaviour in nature; for example, honeybees become more aggressive after consuming ETOH. In previous studies, scientists have used honeybees and fruit flies as models to determine if they showed a strong preference to ETOH. Moreover, ETOH could affect their locomotion and learning abilities. However, whether and how ETOH affects spiders is unclear as of yet. In this study, we used empirical experiments to determine whether spiders showed preference for ETOH, as well as the potential benefits of spiders choosing ETOH, by using a common spider, *Myrmarachne gisti*, which has a high probability of contacting ETOH in their habitat. In our experiment, *M. gisti* showed a significant preference for ETOH. Although the success rate of the first attack was significantly decreased when *M. gisti* were exposed to ETOH, they had a significantly higher predation probability, since fruit flies also showed a significant preference for ETOH. Our findings suggested that ETOH could affect the prey capture efficiency of *M. gisti*, and indicated that spiders might evolve to use ETOH to locate a potential hunting place. Taken together, our findings suggested that *M. gisti* evolved to adapt to ETOH and could use it as a signal of the presence of food resources.

## 1. Introduction

Ethanol (ETOH) is a central nervous system modulator that affects motor skills and learning in both vertebrates and invertebrates [1,2,3]. It is also a natural product which can affect the nervous system without any specific receptor. In the past, its effects on the nervous system had been attributed primarily to nonspecific disturbances in the properties of neuronal membranes [4]. However, some researchers have found that many specific brain proteins are selective to ETOH, such as γ-aminobutyric acid [5], N-methyl-D-aspartate [6], and 5-hydroxytryptamine [7]. The vast majority of the work has been conducted using model systems such as the fruit fly and honeybee [8,9].

Previous studies have focused on *Drosophila melanogaster*, because they have similar behaviour to humans after exposure to ETOH [8]. When fruit flies are exposed to a substantial dose of ETOH, they first stop moving, start to lose postural control, then fall over onto their backs, unable to right themselves, and are finally sedated [8]. It has been proven that both fruit fly adults and larvae have a preference for certain concentrations of ETOH [1,10,11,12]. ETOH can produce many short-term acute behaviors. Fruit flies show a hyperactive phase after exposure to ETOH, which causes them to have a higher speed of movement [13]. In addition, fruit fly larvae can efficiently metabolize ETOH and enhance fitness [14,15]. Another previous study has demonstrated that honeybees became more aggressive after being given 20% ETOH [16]. Moreover, it has also been shown that honeybees experience a significant decrease in their learning ability and ability to discriminate odors following acute exposure to ETOH [2]. The red swamp crayfish (*Procambarus clarkii*) was recently used to study the behavioural effects of ETOH, providing a more generally encompassing view of the effects of ETOH on invertebrates [17]. However, it remains poorly known how ETOH affects carnivorous arthropod predators, such as spiders, which can often contact ETOH in their habitat.

Many previous studies have proven that spiders display odor-based behaviour [18,19,20]. Thus, spiders may also sense and respond to the ETOH. However, previous studies have focused on the effects of neurotoxins, such as scopolamine and caffeine, on the web-building behaviors of some orb web spiders [21,22,23,24]. Others have shed light on how pesticides affect the population quantity and/or sexual behaviour of some non-web-building spiders, including some wolf spiders [25,26]. Very few studies have focused on how ETOH affects spiders. A recent study by Humphrey et al. [27] shows that caffeine could affect the vigilance decrement of a jumping spider. Their findings suggest that drugs which could affect human behavior may also affect that of spiders. It is therefore possible that ETOH, as one of the most common natural products, may have some effects on spiders. Here, we used an ant-mimicking jumping spider, *Myrmarachne gisti* Fox, 1937 (Araneae: Salticidae), to explore the effects of ETOH on spiders, as they have a high probability of contacting ETOH in their own habitat.

*Myrmarachne* is the largest salticid genus, composed of more than 180 described species [28]. *Myrmarachne* is well known for mimicking ants, which is defined as Batesian mimicry. Batesian mimicry provides protection for *Myrmarachne* so they are able to have a wild habitat in the field [29]. *Myrmarachne* occupy a wide range of terrestrial habitats, including rainforests, shrubs, plants, grasses, fields, and even some man-made habitats [30,31,32,33]. *Myrmarachne* can, therefore, come into direct contact with ETOH. On one hand, ETOH is one of the by-products of yeast, resolving sugar even in the presence of oxygen [34,35]. As a result, ETOH is a signal of sugar, which attracts many species of insects [36]. When *Myrmarachne* approached their prey, they had a high probability of coming into direct contact with ETOH, since ETOH would attract their prey. Furthermore, as one of the common flower-visiting arthropods, Jackson et al. [37] have proven that salticids, including *Myrmarachne*, can consume nectar as food, and have a strong preference for artificial nectar. Many studies have proven that ETOH is one of the most common products in floral nectar [36,38,39]. As a result of *Myrmarachne* consuming nectar, they can easily consume ETOH in nature.

In the present study, we investigated whether *M. gisti* has evolved to adapt to ETOH from two perspectives. First, we tested the ETOH preference of *M. gisti*. Then, we determined the potential benefits which *M. gisti* might gain from choosing ETOH. We predicted that: (1) *M. gisti* would show a strong preference for a certain range of concentrations and would avoid other concentrations of ETOH, and (2) they would have a higher predation probability and prey capture efficiency when exposed to the preferred ETOH concentration.

## 2. Materials and Methods

### 2.1. Spider and Fly Maintenance

A total of 640 subadult *M. gisti* (females: *N* = 325; males: *N* = 315, Figure 1A,B) were collected from Wuhan City, Hubei Province, China (30.57° N, 114.33° E, 10 m a.s.l). They were kept individually in glass tubes (height × diameter: 6 × 2 cm) with a 0.5 cm-thick sponge containing water at the bottom of the tube. The glass tubes were plugged with cotton to prevent spiders from escaping. All spiders were under controlled environmental conditions (25 ± 1 °C, 80–85% relative humidity, and a 12:12 h light: dark photoperiod, with the lights turning on at 0800 h), following standard protocols described in other salticid studies [40,41]. We fed spiders with 20 fruit flies (*D. melanogaster*) twice a week and provided water ad libitum until they reached sexual maturity.

The environment of *D. melanogaster* was the same as that of the spiders described above, with autoclaved food used in other *D. melanogaster* studies (5% yeast, 10% dextrose, 7% cornmeal, 0.6% propionic acid, 0.7% agar) [1,42]. The *D. melanogaster* specimens were captured from the wild and raised in the laboratory until they produced offspring. Only the offspring were used in the experiment.

### 2.2. ETOH Preference Test

In order to test whether *M. gisti* could sense the ETOH, we carried out the preference test in a T-maze similar to that of Fischer et al. [42] (Figure 1C). Briefly, the T-maze contained 3 plastic boxes (5 × 5 × 3 cm), with a holding chamber in the middle and two testing chambers on either side. The holding chamber had an introduction hole with a diameter of 3 cm, where spiders would be introduced. The side of the arena was covered with white paper to preclude visual distractions due to their own reflections, given that *M. gisti* has excellent vision [43,44]. The hole was covered with a piece of transparent glass to prevent the spiders from escaping during the experiments. A digital HD video camera (Sony HDR-PJ600E, Tokyo, Japan) was used to record video from the top of the device. Videos were played back after experiments in order to record the time that spiders stayed in each testing chamber.

We placed a 200 μL Eppendorf tube, with little holes at the bottom, at the end of both test chambers. Inside the tube, we placed cotton containing ETOH or water so that the spiders could smell the odor, but not contact the odor resource directly [45]. A total of 10 concentrations (10%, 15%, 20%, 25%, 30%, 35%, 40%, 45%, 50% and 100%) were tested. At the beginning of each trial was a 5-min acclimation phase to remove ownership effects [46], whereby one randomly chosen spider was introduced into the holding chamber, which was separated from the odor boxes by removable iron gauze to prevent the spiders from entering them while the odor was diffusing into the holding chamber. We began the trial by removing the barriers, and last for 5 mins. After each trail, we wiped the start box with 75% ETOH and dried it for 20 min. Then, we swapped the two testing chambers to avoid spatial learning and/or any other cues left by previous spiders [47]. A total of 600 individuals were used in the test. All of the spiders were returned to their individual tubes and rested for at least 7 days before the prey capture experiment. Since ETOH also contains calories, to ensure spiders chose ETOH for its pharmacological properties rather than its caloric value, all used spiders were fed before the experiment.

### 2.3. Predation Probability Test

In order to determine whether *M. gisti* achieved a higher predation probability by choosing ETOH, we carried out a predation probability test by using the same structure as in Section 2.2 (Figure 1C). The pre-treatment method was similar to that described in Fischer et al. [42]. Briefly, flies were wet-starved over 12 h before each trial. This was carried out by placing flies into a clean glass tube with a 0.5 cm thick sponge containing water at the bottom of the tube. Flies over 3 days old were used in the experiments. The concentration of ETOH used in this section was 30%.

At the beginning of each trial, multiple flies (more than 15 flies per trial) were anesthetized with CO_2_. Then, flies were introduced into the holding chamber and given ~2 min to awaken and to remove ownership effects. We lifted the barrier to begin the experiment and replaced the barrier to end the trial when 5 min had elapsed. The number of flies in each testing chamber was recorded for further analysis. A total of 10 groups of fruit flies were tested (a total of 303 individuals were used). After each trial, the whole structure was washed with 75% ETOH and dried for 20 min. The two testing chambers were alternated from one side to the other every trial replicate. All flies were used only once.

### 2.4. Prey Capture Efficiency

We carried out prey capture trials in two plastic boxes (5 × 5 × 3 cm), which were divided into a testing chamber and a holding chamber, in order to examine whether ETOH would change the prey capture efficiency of *M. gisti* (Figure 1D). We placed a 200 μL Eppendorf tube in the end of the testing chamber, which was the same as in the preference experiment. The concentration of ETOH used in this trial was 30%, which was the same as in Section 2.3. The two chambers were separated using filter paper as a barrier to prevent spiders from detecting prey beforehand. Before the trials started, all test spiders were starved for 5 days to ensure they would be motivated to catch prey.

Prey capture trials were preceded by a 5-minute acclimation phase with a spider and two flightless *D. melanogaster* fruit flies (mutant). We lifted the barrier to begin the prey capture phase and replaced the barrier to end the trial when the spider captured the prey or 5 min elapsed, whichever came first. The trial started when the spider’s anterior median eyes oriented towards the prey [48,49]. A total of 82 trials were performed with ETOH (females: *N* = 45; males: *N* = 37) and 70 trials were performed with water as the control (females: *N* = 40; males: *N* = 30). All spiders were randomly chosen and prey were used only once. All spiders were then put back to their habitat after the completion of prey capture efficiency trials.

All prey capture trials were video-recorded from above using the same camera, starting from the acclimation phase to the end of the trial. Videos were played back in order to record the following parameters: time (s) taken from orienting towards the prey to catching it, or a period of 5 min (whichever came first), time (s) that the spider stared at the prey, and the success of the first attack. In nature, spiders usually have one attempt at catching each prey. Thus, we only recorded the success of the first attack.

### 2.5. Data Analysis

All data were checked for normality using the Kolmogorov–Smirnov tests before statistical analyses. Similar to previous papers [6,12,50,51], we calculated a preference index to ensure the preference of ETOH (*PI* 1 and *PI* 2 in *M. gisti* and *D. melanogaster*, respectively) as follows:$$PI\;1=\frac{\mathrm{Time}\;\mathrm{spent}\;\mathrm{in}\;\mathrm{ethanol}\;\mathrm{chamber}-\mathrm{Time}\;\mathrm{spent}\;\mathrm{in}\;\mathrm{water}\;\mathrm{chamber}}{\mathrm{Total}\;\mathrm{time}}$$
$$PI\;2=\frac{\mathrm{Flies}\;\mathrm{in}\;\mathrm{ethanol}\;\mathrm{chamber}\;-\mathrm{Flies}\;\mathrm{in}\;\mathrm{water}\;\mathrm{chamber}}{\mathrm{Total}\;\mathrm{flies}}$$

A positive *PI* indicates an attractiveness, while a negative *PI* represents an avoidance. For the ETOH preference of *M. gisti*, we first used a one-sample *t*-test to confirm whether *PI 1* had a significant difference with 0, then we performed a one-way analysis of variance (ANOVA) followed by planned pairwise comparisons between the relevant groups with a Student–Newman–Keulsa test. For the predation probability test, if the data were normal, we used an independent *t*-test to determine the difference in the number of fruit flies between the ETOH chamber and the water chamber at the end of the test. Otherwise, we performed the Mann–Whitney *U* test. Then, we used a one-sample *t*-test to confirm whether *PI 2* had a significant difference with 0, which indicated the potential hunting place. For prey capture efficiency, if the data were normal, then we used an independent *t*-test. Otherwise, we performed the Mann–Whitney *U* test. Then, a chi-squared test was used to determine the difference in the success rate of the first attack between the ETOH and water groups. All data were analyzed using SPSS Version 25 (IBM SPSS Statistics), and data were graphed using GraphPad Prism 9.

## 3. Results

### 3.1. ETOH Preference Test

*M. gisti* shows attractiveness to the concentrations of 15–35% ETOH and avoidance to all other concentrations (*PI* > 0). Only the concentrations of 20–30% and 45–100% ETOH showed a significant difference with 0, meaning that spiders either were strongly attracted to or avoided ETOH (Figure 2A; Appendix A). Female spiders showed attractiveness at the concentration of 10–30%, but only 20–30% ETOH had a significant attraction. Females showed avoidance to 35–100% ETOH; however, only 45–100% ETOH showed significant avoidance (Figure 2B; Appendix A). As for males, 20–35% ETOH had attraction to spiders, while only 30–35% ETOH had a statistically significant attraction. At the concentrations of 45–100%, there was a significant avoidance (Figure 2C; Appendix A). There were no significant differences among the concentrations of attractiveness or avoidance in both females and males.

### 3.2. Predation Probability Test

The number of fruit flies in the ETOH chamber was significantly higher than that in the water chamber (t = 2.858, df = 18, *p* = 0.01, Table 1). Fruit flies showed a significant preference to 30% ETOH (*PI* = 0.17 ± 0.03, t = 4.758, df = 9, *p* = 0.001).

### 3.3. Prey Capture Efficiency

There was no significant difference in staring time between the ETOH and water groups (*Z* = −0.058, *p* = 0.954; Figure 3A). The capture time between the ETOH and water groups had no significant difference either (*Z* = −1.929, *p* = 0.054; Figure 3B); however, the success rate of the first attack was significantly lower in the ETOH group than in the water group (χ^2^ = 4.131, *p* = 0.042; Figure 3C). In females, the staring and capture time had no statistically significant differences (F = 0.163, t = −0.159, *p* = 0.874 for staring time; F = 0.28, t = 1.301, *p* = 0.198 for capture time; Figure 4A,B). The success rate of the first attack of the ETOH group was significantly lower than that of the water group (χ^2^ = 6.123, *p* = 0.013; Figure 4C). As for males, neither staring time, capture time, nor success rate of first attack had any statistically significant difference (*Z* = −0.247, *p* = 0.805 for staring time; F = 1.197, t = 0.865, *p* = 0.391 for capture time; χ^2^ = 0.105, *p* = 0.746 for success rate; Figure 4A–C).

## 4. Discussion

ETOH can be produced by ripened or rotten plants. The higher the sugar content, the more easily ETOH will be produced. Thus, ETOH is an important stimulus in the environment, and many invertebrates, such as fruit flies and honeybees, show a strong preference for ETOH [9,52,53,54]. Firstly, our results determined that *M. gisti* showed a strong preference for 20–30% ETOH, suggesting that spiders, at least *M. gisti*, could sense and voluntarily choose the ETOH. Then, we tested the potential benefits which spiders may obtain from choosing the specific concentration of ETOH, which we defined as predation probability and prey capture efficiency. Our results show that although spiders achieve a higher probability of predation, they obtain no benefits in prey capture efficiency. Taken together, the data show that *M. gisti* achieves a higher predation probability with a lower prey capture efficiency when exposed to ETOH.

### 4.1. ETOH Preference

Only 20–30% ETOH has a significant attractiveness to spiders, and more than 45% ETOH showed significant avoidance. It has been reported that many insects, such as fruit flies and butterflies, will be attracted by ETOH [8,55]. As a result, the smell of ETOH could be a signal to spiders indicating the location of prey. In nature, ETOH is usually produced in combination with other chemical odors, such as ethyl acetate and isoamyl acetate [34,56]. Compared to other odors, ETOH needs to be orders of magnitude higher in order to be attractive alone. In addition, odors often combine to attract insects. For example, the dwarf palm *Chamaerops bumilis* is attracted by a blend of multiple chemical odors [57]. Therefore, the concentration in our results is relatively higher. Interestingly, the preferred concentration is different between males and females. Males have a slightly higher preferred concentration (30–35%) than females (20–30%); however, there is no statistical significance. Shohat-Ophir et al. [58] found that in *Drosophila*, sexual deprivation will influence ETOH intake. When male *Drosophila* do not mate with females, they tend to intake more ETOH than those who have mated. The male spiders used in our experiment were collected as subadults and monitored until they reached sexual maturity. Therefore, all of the males were virgin, and may have preferred to intake more ETOH, similarly to *Drosophila*. Thus, males show attractiveness to higher concentrations of ETOH. Additionally, the females’ preferred concentration is slightly wider than the males’ preferred concentration. In spiders, females are usually larger than males; therefore, females need more energy to provide for their larger body. In nature, ETOH provides a sugar source for many insects. A wider sense of ETOH suggests more potential prey for females. It is not surprising that both males and females show strong avoidance when the concentration is more than 45%. ETOH is miscible with water in any proportion. High-dose ETOH could take more water from organisms, thus generating irreversible damage or even causing death. On the other hand, previous studies have proven that higher doses will cause reduced movement, loss of postural control, and immobility in *Drosophila* [14]. The higher the concentration, the faster it sedates, which may increase the survival cost of spiders. Therefore, spiders show a preference for ETOH at medium concentrations and avoidance at high concentrations.

### 4.2. Predation Probability and Prey Capture Efficiency

Fruit flies showed a strong preference for 30% ETOH in our experiment. This result suggests that fruit flies tend to gather around the ETOH. It seems that spiders could use ETOH as a signal to locate a potential hunting place, since ETOH will also attract their prey. In the field, non-web-building spiders usually have a relatively low hunting success rate, so they will spend more time searching for prey [59]. It is quite common for spiders to choose a place where they have a higher chance of catching prey. For example, the crab spiders (Thomisidae) usually stay on the flowers in order to prey on the insects visiting inflorescences [60]. ETOH is usually produced by ripe and over-ripe fruit, which could attract many kinds of prey. Thus, it is not surprising that spiders prefer ETOH despite the fact that it decreases their accuracy when catching prey. They will, nevertheless, have a higher probability of catching prey, and could decrease the cost of searching for prey in the field. Our results on the ETOH preference of fruit flies also support this point of view. As a matter of fact, spiders may use odor as a signal for potential food resource; for example, the jumping spider *Evarcha culicivora* prefers the odor of two plants which can provide nectar for them [61].

Surprisingly, our results are contrary to our prediction. The success rate for catching prey on the first attack of the ETOH group was significantly lower than that of the water group. Previous studies [21,62,63,64] have shown that the effects of drugs on spiders are analogous to their effects on humans, so we can hypothesize that ETOH has a similar effect on spiders as it does on humans. When spiders get ‘drunk’, they may become excited and have faster speeds of movement; however, the accuracy will be decreased, meaning the spiders cannot catch the prey precisely. Another study involving *Drosophila* showed that although chronic ETOH exposure will lead to males’ hypersexuality, causing them to chase anything with wings, including other males, the success rate of mating will be lower [65]. This means that the accuracy will be decreased in males when *Drosophila* consume ETOH. This result is similar to our results in terms of prey capture time and staring time. The spiders in the ETOH group had a longer mean time of catching prey than those in the water group (ETOH group: 70.71 s; water group: 52.49 s), although the difference was not statistically significant (*p* = 0.054). Additionally, the spiders in the ETOH group had a slightly longer mean staring time than those in the water group (ETOH group: 6.16 s; water group: 5.92 s); however, the data were not significant. Spiders seemed to spend more time locating and confirming the prey before attacking. The accuracy of catching the prey precisely decreased, similarly to the decreased mating success rate in *Drosophila*. However, due to the fact that the prey will be attracted by the ETOH, the lower prey capture efficiency could be balanced by the greater amount of prey available. In fact, when we focus on the success rate of catching prey, the rate of the ETOH group was slightly higher than that of the water group, although there was no statistically significant difference (Chi-square test, ETOH: 88.06%, water: 75.61%, χ^2^ = 3.745, *p* = 0.053). Thus, spiders will choose and be attracted to ETOH.

Additionally, the success rate for catching prey on the first attack between the two groups of males had no significant difference, while there was a statistically significant difference between the two female groups. This result indicates that females are more sensitive to ETOH than males. According to our results for ETOH preference, females showed a significant attractiveness to doses between 20–30%, while males showed a significant attractiveness to doses between 30–35%. In our research, we used 30% ETOH; this concentration is at the higher end of the female preference range and the lower end of that for males. Thus, at this concentration, females may sedate faster than males, which shows a higher sensitivity to ETOH.

In our study, we found that *M. gisti* do not obtain any benefits for prey capture efficiency after exposure to ETOH; however, spiders may have a higher probability to catch prey, as fruit flies will also be attracted to ETOH. In nature, prey is usually concentrated around ripened/rotten fruits. When they come to feed, they consume ETOH, which can affect their behavior (e.g., slower, more careless, etc.). Then, spiders use the ETOH odor in order to find such a hunting place, and start to hunt their prey immediately. This means the time of spiders’ direct contact with ETOH may be far shorter than that of prey. In the future, we should test the difference capture efficiencies between long and short exposure to ETOH in spiders. Spiders may obtain other benefits from contacting ETOH as well. It has been reported that *Drosophila* use ETOH to protect larvae from certain parasitoid wasp species [66]. In addition, ETOH will kill other bacteria, which could provide a relatively safe environment [36]. In the future, we should test the survival rate, the development time, and the fecundity under long-term exposure to ETOH. In addition, it has been reported that many insects (such as fruit flies and honeybees) have evolved to have a tolerance to ETOH due to contact with it [2,67,68]. Bainton et al. [1] have proven that dopamine plays a role in the response of *Drosophila* to ETOH. Similarly, Mustard et al. [69] have found that honeybees will preferentially consume sucrose solutions that contain ETOH, although high doses of ETOH will increase their mortality rate. More work is needed to find out whether *M. gisti* have evolved a similar tolerance, as well as the other reasons for choosing a specific concentration of ETOH, e.g., for self-satisfaction, similarly to humans. Moreover, there are over 130 different families of spiders in the world living in various habitats [28]; we should also test other types of spiders, such as other non-web-building and web-building spiders.

In summary, we found that an ant-like jumping spider, *M. gisti,* showed a preference for ETOH. When exposed to ETOH, *M. gisti* demonstrated lower prey capture efficiency; however, their prey capture probability will still be higher, due to fruit flies being attracted by ETOH. Our findings suggest that spiders may evolve to use ethanol as a signal for the presence of food resources, and provide a new direction for ETOH–invertebrates interactions. In order to understand the underlying mechanism of ETOH effects, as well as how ETOH affects invertebrates more generally, future studies are needed to identify other short- and long-term effects and molecular mechanisms.

## Figures and Tables

**Figure 1 insects-13-01009-f001:**
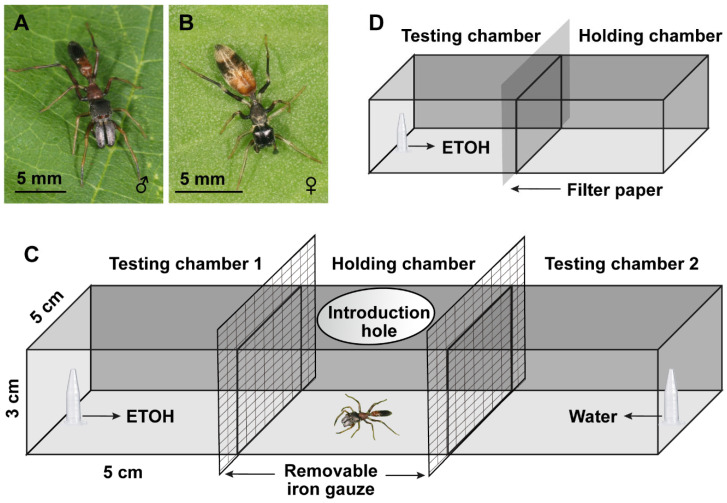
The (**A**) adult male and (**B**) adult female ant-mimic jumping spider, *Myrmarachne gisti*, and the experimental structure of the (**C**) preference and predation probability tests and (**D**) prey capture efficiency test. The dimension of (**C**,**D**) are not scaled.

**Figure 2 insects-13-01009-f002:**
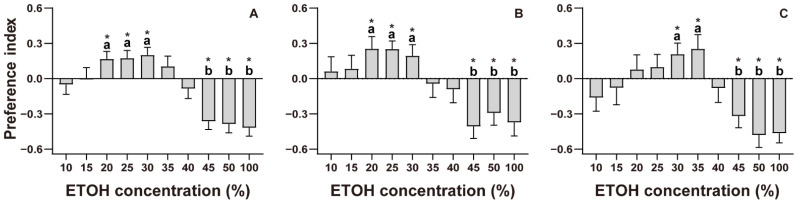
Ethanol (ETOH) preference of male and female *Myrmarachne gisti,* showing (**A**) preference index (*PI*) for both genders (females: *N* = 325; males: *N* = 315), (**B**) *PI* for females, and (**C**) *PI* for males. ‘*****’ indicates a significant difference from 0 (*p* < 0.05). One-way analysis of variance (ANOVA) and Student–Newman–Keulsa tests were used for multiple comparisons. Significant differences between the groups are indicated with letters.

**Figure 3 insects-13-01009-f003:**
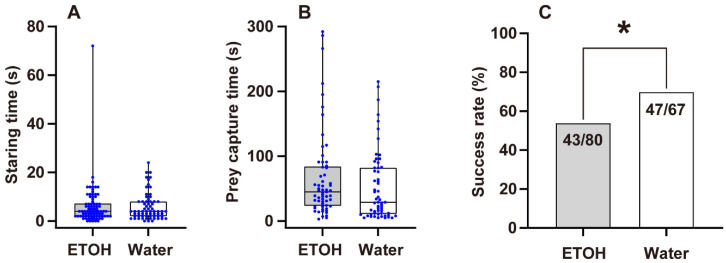
Prey capture efficiency of *Myrmarachne gisti*, showing (**A**) the time spiders spent staring at prey (females: *N* = 85; males: *N* = 67), (**B**) the time taken to capture prey (females: *N* = 68; males: *N* = 53), and (**C**) the success rate of their first attack. Boxplots show the data as jittered dots, with the box indicating the interquartile range (IQR), the whiskers showing the range of values that are within 1.5 × IQR, and a horizontal line indicating the median. ‘*****’ indicates a significant difference (*p* < 0.05).

**Figure 4 insects-13-01009-f004:**
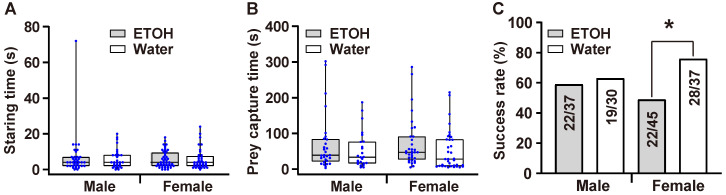
Prey capture efficiency of *Myrmarachne gisti* for males and females separately, showing (**A**) the time spiders spent staring at prey, (**B**) the time taken to capture prey, and (**C**) the success rate of their first attack. Boxplots show the data as jittered dots, with the box indicating the interquartile range (IQR), the whiskers showing the range of values that are within 1.5 × IQR, and a horizontal line indicating the median. ‘*****’ indicates a significant difference (*p* < 0.05).

**Table 1 insects-13-01009-t001:** The numbers of fruit flies in the ETOH chamber and water chamber at the end of the experiment.

Group Number	ETOH	Water
G1	9	5
G2	15	16
G3	8	8
G4	13	6
G5	19	8
G6	11	4
G7	9	3
G8	6	3
G9	9	2
G10	10	3

## Data Availability

The data presented in this study are available upon request from the corresponding author.

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
