# Peer review of "An Ant-Mimicking Jumping Spider Achieves Higher Predation Probability with Lower Success Rate When Exposed to Ethanol"

_insects, 2022, doi:10.3390/insects13111009_

Round 1

Reviewer 1 Report

I read it carefully and I can claim the manuscript is very interesting, inspiring and valuable.

1) I noticed only one mistype - line 110 (testing chamber), but the generic name of spider (Myrmarachne) is mutilated everywhere except Abstract - Myrmrachne/Mymachne.... Correct it, please.

2) I think, Tabs 1-3 are redundant - all information are visible from Fig. 2.

3) Sentence ad lines 315-316 must be deleted - it is against scientifical rules. If you have a hypothesis about a larger experimental set and better statistical results, you must test it de novo, it is not possible to increase (step-by-step) your sample controlling changes in significance. Such action is common(?) but strongly non-scientific.

4) I think, the discussion about results can be improved by careful evaluating of strict timing aspects of methods. Try to follow me: in nature, there are a lot of fruitflies about rotting fruits. They consume ETOH and they can be affected in their behaviour (slower, careless etc.). Spiders can use the ETON odour to find such fruit and immediately prey on drunken flies. It means - your 5 minutes acclimation phase can be too short for flies to be slowed and too long for spiders (as they are drunker than in natural conditions). It means - your results are valuable, but strongly influenced by the (artificial, non-discussed, unjustified) timing of the experiment. I can hypothesise, a longer exposition of flies and a shorter exposition of spiders can increase their capture efficiency. You should discuss this aspect too.
Nevertheless, your Discussion is very nice, I found it brilliant, but I miss this topic.

Reviewer 2 Report

Thank you for sending your manuscript for review. Your chosen topic is very interesting, I like the methodology and the results discussed. I have a few comments on the paper, but they are mostly minor. I also have a few questions for you about your experiment. Don't take this as a criticism, it's more about clarifying ambiguities or topics to think about. I would be happy if you continue similar research.

First, then, on what I think should be modified.
In your introduction you use both common names and Latin names of animals. In scientific publications, usually the first mention of the species in question will give both names, if they exist, and then one of them is used. Or one variant can be chosen, but it should be consistent within the text.

In the third and fourth paragraphs of the introduction you have a repeated typo in the word "Myrmarachne".

The methodology section details how the spiders were obtained and kept. For the fruit flies, you state that they were bred in the same environment, but you do not state where they came from. Were they individuals captured from the wild or raised in human care?

The individual experiments are well described, including illustrations of the testing apparatus, which I very much appreciate. However, what is missing is a summary of what the overall procedure was. Specifically, how many individuals entered each phase of the experiment and whether any were discarded. The sentence at the end of each paragraph that each animal was only used once is misleading. I assume it was meant to be used for a specific part of the experiment, since you state that the spiders were further used for the prey-capture experiment. For example, the numbers of individuals in the first experiment can only be found in the results. It would also be useful to indicate which individuals were selected for the final test, whether this was related to the distribution in the first test with different concentrations of ETOH or whether they were selected randomly, as with the control group.

The details of the camera used are repeated, but I am more bothered by the fact that more detailed information about the type of camera is not given until the last paragraph. It is better to give the full information in the first paragraph, and later just state that it is the same camera.

The results are written quite briefly, but accompanied by plenty of tables and graphs with detailed values. The only problem I have is with the graph in Figure 3, which seems redundant. All the information is summarized in the first sentence. I understand that a paragraph with one sentence does not work well, but I would consider the necessity of this chart. For example, a table could be made with the numbers of fruit flies that were in the ethanol chamber and the water chamber at the end of the experiment. It wouldn't even be too extensive for the ten groups tested.
I have another question about this test, was it always possible to divide the fruit flies into these two parts? I mean, according to the drawing of the apparatus, did it include the holding chamber, were any of the fruit flies in this compartment at the end of the test, or were they included in the results?

I also have some additional questions, it is up to the authors to add some of these topics to the manuscript.
First, I am interested in the reasons why this particular species was chosen. There are several jumping spider that may encounter ethanol in their environment. Moreover, it is a mimetic species. What were the criteria for selecting M. gisti?

In your calculations, you take into account the success of the first attack. Do you also have records of how many individuals caught prey and how many were terminated without successful capture?
If the presence of ethanol increases the chances of finding a suitable place to hunt, it might not be a big problem if the first attack is unsuccessful because the prey is more concentrated in such a place. Thus, the lower hunting efficiency due to ethanol would be balanced by the greater amount of prey available. (Think of it as food for thought.)

You state that some concentrations of ethanol are avoided by spiders. I can imagine this happening at higher concentrations that might cause problems for the spider. However, lower concentrations may not be avoidance, but those concentrations are not very attractive. From the records taken, it would be possible to compare behavior at low vs. high concentrations that are not significantly attractive to spiders. Would there be a difference?
